# Hyperthermic Intraperitoneal Chemotherapy: A Critical Review

**DOI:** 10.3390/cancers13133114

**Published:** 2021-06-22

**Authors:** Wim Ceelen, Jesse Demuytere, Ignace de Hingh

**Affiliations:** 1Department of GI Surgery, Ghent University Hospital, 9000 Ghent, Belgium; jesse.demuytere@ugent.be; 2Cancer Research Institute Ghent (CRIG), 9000 Ghent, Belgium; 3Department of Surgery, Catharina Cancer Institute, PO Box 1350, 5602 ZA Eindhoven, The Netherlands; ignace.d.hingh@catharinaziekenhuis.nl; 4GROW—School for Oncology and Developmental Biology, Maastricht University, PO Box 616, 6200 MD Maastricht, The Netherlands

**Keywords:** peritoneal, HIPEC, intraperitoneal, drug transport

## Abstract

**Simple Summary:**

Patients with cancer of the digestive system or ovarian cancer are at risk of developing peritoneal metastases (PM). In some patients with PM, surgery followed by intraperitoneal (IP) chemotherapy has emerged as a valid treatment option. The addition of hyperthermia is thought to further enhance the efficacy of IP chemotherapy. However, the results of recent clinical trials in large bowel cancer have put into question the use of hyperthermic intraperitoneal chemotherapy (HIPEC). Here, we review the rationale and current results of HIPEC for PM and propose a roadmap to further progress.

**Abstract:**

With increasing awareness amongst physicians and improved radiological imaging techniques, the peritoneal cavity is increasingly recognized as an important metastatic site in various malignancies. Prognosis of these patients is usually poor as traditional treatment including surgical resection or systemic treatment is relatively ineffective. Intraperitoneal delivery of chemotherapeutic agents is thought to be an attractive alternative as this results in high tumor tissue concentrations with limited systemic exposure. The addition of hyperthermia aims to potentiate the anti-tumor effects of chemotherapy, resulting in the concept of heated intraperitoneal chemotherapy (HIPEC) for the treatment of peritoneal metastases as it was developed about 3 decades ago. With increasing experience, HIPEC has become a safe and accepted treatment offered in many centers around the world. However, standardization of the technique has been poor and results from clinical trials have been equivocal. As a result, the true value of HIPEC in the treatment of peritoneal metastases remains a matter of debate. The current review aims to provide a critical overview of the theoretical concept and preclinical and clinical study results, to outline areas of persisting uncertainty, and to propose a framework to better define the role of HIPEC in the treatment of peritoneal malignancies.

## 1. Introduction

Peritoneal metastases (PM) are a common manifestation of abdominal malignancies, most frequently occurring in patients with upper gastrointestinal, colorectal, and ovarian cancer [1,2,3]. Although less often, primary solid tumors outside the peritoneal cavity such as malignant melanoma, lung cancer, and lobular breast cancer may also metastasize to the peritoneum [4,5]. An increased awareness amongst physicians as well as the improvement of radiological techniques such as diffusion-weighted MRI have resulted in an increasing incidence of PM being reported in population-based studies in recent years. When taking all the origins together, PM pose a significant burden on current oncological care.

For long, it has been recognized that systemic treatment of PM appears to be less effective as compared to lung or liver metastases [6]. Poor vascularization of the peritoneal cavity may play a role, but the exact mechanisms underlying this phenomenon remain to be elucidated. As anticancer drugs are usually administered systemically exposing healthy tissue, their therapeutic index is limited. Some of these shortcomings can be addressed by local or locoregional delivery of chemotherapy. During this mode of anticancer therapy, drug is administered either through a feeding artery, or into an anatomical cavity. Locoregional drug delivery allows to administer a higher dose with less systemic toxicity. Examples include hepatic artery infusion and instillation in the peritoneum (intraperitoneal, IP), bladder (intravesical), brain ventricles (intrathecal), and chest cavity (intrapleural).

Intraperitoneal chemotherapy takes advantage of the large surface area of the peritoneum (approximately 2 m^2^) to enable mass transfer either from the peritoneal cavity to the systemic circulation (drug therapy), or vice versa (dialysis). The origins of the peritoneal route of drug delivery can be traced back to the eighteenth century: in 1744, the English surgeon Christopher Warrick, instilled a mixture of ‘Bristol water’ and Bordeaux wine in the peritoneal cavity of a patient with intractable ascites, apparently with great success [7]. There was some enthusiasm during the first half of the twentieth century for IP administration of radioactive gold (^198^Au) in the adjuvant and palliative treatment of ovarian cancer, but significant morbidity was observed [8]. Also, intraperitoneal radioactive chromic phosphate (^32^P) administration was attempted for ovarian cancer, but this led to significant complications and resulted in inhomogeneous drug distribution [9].

The interest in intraperitoneal drug delivery (IPDD) was rekindled with the publications of Dedrick in the 1970s. He proposed a theoretical framework for IPDD based on the pharmacokinetic (PK) advantage that results from the fact that systemic drug clearance is much faster compared to peritoneal clearance. As a result, IP drug can be administered at a higher dose with low systemic exposure and toxicity [10]. Of note, Dedrick was also one of the first authors to emphasize that despite the obvious PK advantage of IPDD, the resulting tissue penetration depth is very limited [11].

The use of hyperthermia to treat cancerous growths dates from several millennia ago and continues to find applications in modern medicine. The concept of combining IPDD with hyperthermia as a hyperthermic IP chemoperfusion (HIPEC) was first studied in an animal model in 1974 by Euler [12]. The first clinical use of HIPEC was reported in 1980 by Spratt et al., who performed hyperthermic chemoperfusion with thiotepa in a patient with pseudomyxoma peritonei (PMP) [13].

In the following decades, HIPEC was introduced in the treatment of peritoneal metastases from a variety of primary malignancies and in primary peritoneal malignancies including peritoneal mesothelioma. Long surrounded by skepticism, HIPEC is now offered at hundreds of treatment centers worldwide [14]. Nevertheless, the efficacy and safety of HIPEC remain debated and hamper the universal acceptance by the oncology community. Proponents will argue that the addition of HIPEC was recently shown to prolong survival in ovarian cancer in a randomized clinical trial (RCT) but criticism was undoubtedly fueled by negative results of RCTs in patients with colorectal cancer (CRC) PM [15].

The aims of this review are to provide a critical overview of the theoretical concept and preclinical and clinical study results, to outline areas of persisting uncertainty, and to propose a framework to better define the role of HIPEC in the treatment of peritoneal malignancies.

## 2. Basic Concepts

### 2.1. Pharmacokinetic Behavior and Drug Tissue Transport after Intraperitoneal Chemotherapy

The pharmacokinetic rationale for IPDD is based mainly on the presence of the peritoneal-plasma barrier [16]. The resulting (PK) advantage can be expressed as the parameter R_d_, calculated as (C_P_/C_B_)IP/(C_P_/C_B_)IV where C_P_ and C_B_ represent the peritoneal and blood concentrations, respectively (Figure 1). The pharmacokinetics of IPDD is usually described using compartmental models, which consist of a systemic and a peritoneal compartment. These idealized compartments are separated by the peritoneal-plasma barrier, which is characterized by a permeability-area (PA) product. Based on experimental correlations of measured drug clearance with molecular properties, it was estimated that the PA is inversely proportional to the square root of the molecular weight of the drug [17].

Intraperitoneal drug delivery allows to reach a high IP drug concentration. However, the anticancer efficacy depends on the tissue drug concentration. Therefore, the extent of mass transport into the tissue is an essential parameter that determines the efficacy of IPDD. Simulations and experimental studies consider tumor tissue as a homogeneous (isotropic) porous medium. Two major mechanisms determine the transport of drug into tumor tissue: convection or bulk fluid flow, which is driven by a pressure gradient, and diffusion, resulting from a concentration gradient. In reality, both mechanisms occur simultaneously, and the ratio of convective over diffusive transport is quantified as the Péclet number. The Péclet number is low for small molecules (diffusion dominates) and higher for large compounds such as antibodies or nanoparticles, for which tissue penetration mainly depends on a pressure difference.

#### 2.1.1. Convection

During HIPEC, the extent of convective drug transport is proportional to the difference in pressure between the fluid filled peritoneal cavity, and the stromal tissue pressure. Since chemotherapeutics will interact with cellular and stromal structures, the velocity of the compound is always slower than that of the carrier fluid in which it is dissolved. The ratio of both velocities is termed the retardation or hindrance coefficient. The hydrostatic pressure exerted by the intraperitoneal fluid column can be estimated as 10–20 cm H_2_O (7.4–14.8 mm Hg). The tissue pressure that resides in the peritoneal cancer tissue has never been measured clinically, but results from preclinical experiments and numerical simulations suggest that it is much higher compared to normal tissue. First, tumor tissue is characterized by an elevated interstitial fluid pressure (IFP) caused by increased blood flow, ‘leaky’ capillaries, and deficient lymphatic drainage [18]. Second, tumor tissue is associated with elevated solid stress, arising from different sources [19]. External solid stress is exerted on the tumor as it grows and compresses the surrounding tissue. Swelling solid tissue stress is the result of electrostatic repulsion between negatively charged stromal components such as hyaluronic acid [20]. A third component is residual solid tissue stress, which represents the stored elastic energy which can be observed when cutting into a solid tumor: this leads to its bulging and expansion. Pressure driven (convective) drug transport also depends on the hydraulic conductivity of the tissue, which is affected by the viscosity of the interstitial fluid and by mechanical stiffness of the tumor stroma [21].

#### 2.1.2. Diffusion

According to Fick’s law, diffusive mass transport is driven by a concentration gradient. In addition, the rate of drug diffusion depends on temperature, physicochemical drug properties, and on the stromal architecture [22]. Relevant drug properties include its molecular weight, hydrodynamic size, charge, and configuration. Important properties of the stroma or extracellular matrix (ECM) that affect drug diffusion are cellular composition, density, stiffness, visco-elasticity, and geometrical fiber arrangement [23]. In cancer tissue, there is increased deposition of collagen I, resulting in increased stiffness or rigidity compared to normal tissue. Also, tumors overexpress the collagen cross-linking enzyme lysyl oxidase (LOX), which further increases stromal stiffness [24]. Also, the geometric arrangement of collagen fibers affects drug diffusion: experimental studies show that fibers that are oriented tangentially from the tumor surface direct drug diffusion away from the tumor, while the opposite occurs when fibers are radially aligned [25].

### 2.2. Penetration Depth after IPDD

An important limitation of IPDD is the very limited penetration distance in tumor tissue, which is a few millimeters at most, depending on drug, treatment, and tissue properties [26]. This is explained by the elevated pressure characterizing the biophysical TME, and by the very low hydraulic conductivity of tumor tissue, which is typically in the range of 10^−15^–10^−14^ m^2^/pa·s in colorectal PM as measured using modified Ussing chambers. Only limited clinical data are available on tissue penetration after IPDD. Several authors have reported drug concentrations in tissue homogenates after HIPEC, but this is a poor substitute for the actual penetration distance. Preliminary data from a study comparing normothermic versus hyperthermic chemoperfusion with cisplatin for ovarian cancer (NCT02567253) show that platinum penetrates normal stroma much easier than the cancer nodules (Figure 2). As a consequence, numerous physical, chemical, and pharmacological approaches have been tested preclinically in order to enhance drug penetration after IPDD, and the interested reader is referred to a recent review on this topic [27]. In clinical trials, several approaches are being tested that target matrix deposition, matrix remodeling, and cell-matrix interactions in solid cancers [28]. However, none of these trials use IPDD for PM.

### 2.3. Use of Hyperthermia

The use of hyperthermia in oncology has a long history and is based on several parallel observations. First, the use of hyperthermia is selectively lethal for malignant cells [29]. Second, the effects of heat can be synergistic with those of other anticancer treatments, including chemotherapy and radiotherapy [30]. There is considerable heterogeneity in the extent, timing, and underlying mechanisms of thermal enhancement of chemotherapy. Synergism with heat is particularly evident for the platinum compounds and mitomycin C. However, other agents such as the taxanes and the antimetabolites do not show thermal enhancement. Third, hyperthermia improves tissue perfusion and oxygenation, and may enhance tissue penetration. In a rodent model of colorectal peritoneal cancer, Los and coworkers found significantly higher tumor platinum concentrations when IP cisplatin was combined with regional hyperthermia (41.5 °C) [31]. Other drugs showing increased tumor penetration when combined with hyperthermia include carboplatin, oxaliplatin, and doxorubicin [32,33]. Of note, many of the in vitro studies that have aimed to establish thermal enhancement of chemotherapy have used temperatures, exposure times, and drug concentrations that are not clinically relevant or achievable. Helderman and coworkers recently performed a series of in vitro experiments using clinically relevant conditions (38–43 °C for 60 min) in several 2D and 3D human colorectal cancer cultures [34]. They showed that thermal enhancement of cytotoxicity is highly dependent on the cell line and on the drug used: thermal enhancement was evident for oxaliplatin and cisplatin, but not for mitomycin C, carboplatin, or 5-FU. Interestingly, hyperthermia may diminish the systemic toxicity of some drugs, such as doxorubicin and cyclophosphamide by increasing their alkylation and/or excretion [35]. Besides the choice of drug, also length of the exposure to hyperthermia might play a crucial role. This was recently investigated using patient-derived organoids from colorectal cancer PM. In this study by Forsythe, low dose heated oxaliplatin (200 mg/m^2^) for 200 min appeared to be more effective in terms of cytotoxicity than a higher dose of oxaliplatin (460 mg/m^2^) for only 30 min [36].

The ideal target temperature of HIPEC is unknown. In vitro, DNA repair is inhibited at a temperature >41 °C, but the relationship between temperature and anticancer efficacy in vivo is not known. Also, due to the heat sink effect of the tumor blood vessels, the actual tissue temperature that can be reached is lower than that of the heated IP solution. Hyperthermia elicits the expression of heat shock proteins (HSP’s), which were shown to exert anti-apoptotic and proliferative effects, and to induce resistance to chemotherapy [37,38]. Also, temperatures above 41 °C may cause scald injury to the peritoneum, which is already extensively damaged by the CRS [39].

There is only one clinical study in the ‘prophylactic’ setting that has compared normothermic with hyperthermic chemoperfusion. Yonemura and coworkers randomly allocated patients with T2-4 gastric cancer without peritoneal metastases who underwent gastrectomy with extended lymphadenectomy to either surgery alone (*N* = 47), surgery with HIPEC (mitomycin C and cisplatin, *N* = 48), or surgery with normothermic chemoperfusion (*N* = 44) [40]. In univariate analysis, overall survival was better in the hyperthermic group compared to the two other groups. In a multivariable (Cox) model, the hazard ratio for death of normothermic versus hyperthermic chemoperfusion was 1.77 (95% confidence interval 0.91–3.42, *P* = 0.092). The methodological quality of this randomized trial (according to the CONSORT guidelines) was, however, moderate.

In the era of immune therapy, there is renewed interest in the potential immune stimulating effects of hyperthermia [41]. Hyperthermia leads to immunogenic cell death by secretion of damage associated molecular patterns (DAMPs) including calreticulin, ATP, high mobility group B1 (HMGB1), and heat shock proteins 90 and 70. These patterns may activate antigen presenting cells and mobilize an effective T cell mediated immune response. Also, hyperthermia may reverse the ‘cold’ tumor microenvironment (TME) observed in most PM to a highly immunogenic TME, which sensitizes tumors to immune checkpoint inhibition. In studies that combine radiotherapy with external hyperthermia, immune modulating effects were observed, leading to an abscopal therapy response [42]. Others have shown, however, potentially adverse effects of local heating on overall tumor immunity [43]. HSPs s can promote cancer growth and malignant behavior by the induction of extracellular matrix remodeling, resistance to apoptosis, epithelial to mesenchymal transition, tumor angiogenesis, and metastasis [44]. A recent phase-2 clinical trial investigated the potential additional value of autologous tumor antigen-loaded αDC1 vaccine in patients undergoing CRS and HIPEC for peritoneal metastases. The therapy appeared to be well tolerated by patients but the effect of vaccination on median survival appeared to be limited and it was concluded that this was not a good strategy to pursue [45].

Very little is known on the effect of HIPEC on the TME of PM, and on the peritoneal immune environment. Franko and coworkers sampled peritoneal fluid during HIPEC procedures at different time intervals between 0 and 90 min [46] They did not observe significant changes in the number of peritoneal NK cells, CD4/CD8 ratio, or granulocyte/lymphocyte ratio during the course of HIPEC.

## 3. Clinical Implementation of Hyperthermic Intraperitoneal Drug Delivery

The basic setup used for HIPEC treatment consists of one or more inflow- and outflow tubes and temperature probes, one or more roller pumps, and a heating element. Several HIPEC devices are commercially available. There is considerable heterogeneity in the procedural parameters that are used to administer HIPEC: drug type and dose regimen, carrier solution, target temperature, treatment duration, and delivery technique all vary substantially according to local preference [47]. As a result, many different HIPEC-regimens are currently used and standardization is sparse, hampering pooling of outcome data [48].

### 3.1. Choice and Combination of Chemotherapy

Ideally, chemotherapy drugs for HIPEC should have the following properties: a favorable pharmacokinetic profile, no cell cycle specificity, and absence of local peritoneal toxicity. Unfortunately, all chemotherapeutics currently administered during HIPEC are used off label. In colorectal cancer, debate persists on the use of oxaliplatin versus mitomycin C for HIPEC. Results from retrospective studies are difficult to interpret due to differences in clinical and treatment parameters [49]. A prospective randomized trial in appendiceal cancer showed that compared to mitomycin C, the use of oxaliplatin for HIPEC was associated with a better safety and quality of life profile [50,51]. However, oxaliplatin as a HIPEC agent failed in recent randomized trials in colorectal cancer. Possibly, additional factors such as choice of carrier solution, target temperature, and treatment duration are important determinants of the efficacy of oxaliplatin, as recently demonstrated in organoid models [36,52].

Although it seems intuitively appealing to combine drugs for HIPEC, several caveats should be taken into consideration. First, unsuspected chemical or physical incompatibilities may exist that preclude the administration of two or more drugs IP in the same solution. Second, when toxicity occurs, it will be problematic to find out which agent is responsible for which observed toxicity. Third, prospective clinical trials do not support the use of multi-agent HIPEC regimens. Quénet and coworkers showed that, compared to HIPEC with oxaliplatin alone, the addition of irinotecan significantly increased the complication rate, but did not benefit recurrence-free or overall survival [53].

### 3.2. Open Versus Closed Abdomen Perfusion

Chemoperfusion with the skin and/or fascial layer closed theoretically prevents contamination of the OR environment and heat loss and may enhance convection driven tumor chemotherapy penetration due to increased IP pressure. The open technique (‘coliseum’), on the other hand, allows to manually stir the abdominal contents in order to ensure homogeneous drug and temperature distribution. Prospective comparative studies are lacking, but retrospective data suggest that both techniques are comparable in terms of intraoperative hemodynamics and postoperative morbidity [54,55]. Recent developments include the use of CO_2_ recirculation and laparoscopy assisted HIPEC [56,57].

## 4. Clinical Results of HIPEC

The results of the most important randomized clinical trials that have investigated HIPEC are summarized in Table 1.

### 4.1. Ovarian Cancer

The majority of epithelial ovarian cancer (EOC) patients presents with peritoneal metastases and around 75% will relapse in the peritoneal cavity after successful first line treatment. Therefore, EOC appears to be the ideal candidate for IPDD and remains the best studied indication. In addition, serous primary peritoneal cancer is regarded as sharing many molecular and clinical features with EOC and is treated similarly [65]. The multicenter randomized OVHIPEC trial investigated the additional benefit of cisplatin based HIPEC to cytoreductive surgery (CRS) after neoadjuvant chemotherapy (NACT) in patients with primary EOC who were initially not eligible for CRS due to extensive peritoneal involvement. It was found that the addition of HIPEC to interval CRS resulted in a significantly better progression free survival, while overall survival increased from 33.9 to 45.7 months [63]. Addition of HIPEC in these patients did not result in more postoperative complications, did not negatively affect the quality of life and appeared to be cost-effective [66,67]. Based on these results, the current guidelines of the US National Comprehensive Cancer Network (NCCN) indicate that HIPEC can be considered for all patients with stage III EOC for whom NACT and interval CRS is performed [68,69]. A second smaller trial, published as abstract only, used a lower IP cisplatin dose, included primary as well as interval CRS patients, and failed to demonstrate a significant difference in PFS [70]. In recurrent EOC, the results of a small RCT showed a superior overall survival after CRS and HIPEC versus surgery alone (26.7 versus 13.4 months, *P* = 0.006) [62]. However, the methodological quality of that trial was only moderate. A recent randomized phase II trial assigned recurrent EOC patients to either CRS and HIPEC (Carboplatin 800 mg/m^2^ for 90 min) or CRS alone, followed by five or six cycles of postoperative IV carboplatin-based chemotherapy, respectively [64]. Based on a ‘pick-the-winner’ design, an arm would be considered superior if at least 17 out of 49 patients were without progression at 24 months follow-up. The results showed that at 24 months, eight patients (16.3%) in the HIPEC arm and 12 (24.5%) in the CRS alone arm were free of progression, indicating that the addition of HIPEC with Carboplatin did not lead to superior outcomes.

Currently, the international OVHIPEC-2 consortium is investigating the role of HIPEC in patients with FIGO stage III ovarian cancer who are treated with primary CRS. In total, 538 patients will be randomized to CRS with or without HIPEC [71]. A French multicenter randomized trial (CHIPOR; ClinicalTrials.gov identifier: NCT01376752) was initiated in 2011 and compares CRS alone with CRS plus HIPEC in patients with recurrent ovarian cancer. Interestingly, the recently initiated Australian HyNOVA (Hyperthermic versus Normothermic intraperitoneal chemotherapy following interval cytoreductive surgery for stage III epithelial OVArian cancer) will compare interval debulking and chemoperfusion with cisplatin (100 mg/m^2^, 90 min) at two different temperatures: 42.0 °C and 37.0 °C [72]. Currently, therefore, the available evidence supports the use of HIPEC in association with interval CRS in stage III EOC. The results of ongoing trials are awaited to establish its role in primary CRS and recurrent cases.

### 4.2. Colorectal Cancer

In 2003, a randomized clinical trial showed that CRS and HIPEC (90 min, mitomycin-C 35 mg/m^2^) improved survival in patients with colorectal peritoneal metastases as compared to palliative surgery and systemic treatment alone (22 versus 12 months respectively) [73]. Ever since, numerous non-controlled studies have shown that long-term survival can be obtained with CRS and HIPEC with median survivals ranging from 14.6 to 60.1 months according to a recent review [74]. Although the treatment-related mortality in the trial by Verwaal was as high a 8%, this has decreased significantly with increasing experience and is currently as low as 1–2% in most centers [75].

A recently published French multicentre study compared CRS alone with CRS combined with short duration (30 minutes) oxaliplatin (460 mg/m^2^) based HIPEC in colorectal PM (PRODIGE 7/ACCORD 15, NCT00769405) [15]. Interestingly, the addition of HIPEC did not improve OS in this trial, but did increase 90 day morbidity. This raises the question concerning the value of HIPEC in addition to complete CRS in colorectal cancer PM. A possible explanation for the lack of efficacy of the oxaliplatin-based regimen in the PRODIGE 7 trial may be the selection of patients as these were only included after a minimum of 6 months of systemic therapy. Such therapy was oxaliplatin-based in the majority of patients and this may have resulted in an acquired resistance of the peritoneal metastases against IP oxaliplatin as was recently demonstrated in a pre-clinical study [76]. Also, patient-derived organoids from colorectal peritoneal metastases appear to be resistant to heated oxaliplatin in a dosage similar to the one used in the PRODIGE7 protocol [77].

Another topic of debate is whether systemic treatment, either neo-adjuvant, adjuvant or both should be part of the initial treatment strategy. Although peri-operative treatment was part of the PRODIGE7 study protocol and is practised widely around the world, high-level evidence to support this practice is currently lacking [78,79]. In a recent retrospective comparative cohort study, no beneficial effect of peri-operative systemic therapy was shown after complete CRS and HIPEC [80]. In contrast, a large population based study including 393 patients undergoing CRS and HIPEC revealed a benefit of adjuvant systemic treatment as compared to standard follow up alone after propensity score matching. The value of perioperative chemotherapy is currently investigated in the international multicenter randomized CAIRO6-trial [81].

Besides a role for HIPEC in the treatment of established PM, also the role of ‘prophylactic’ two randomized trials have evaluated the use of HIPEC with oxaliplatin in patients at high risk of peritoneal recurrence (i.e., perforated tumors, pT4 tumors, minimal PM resected at the time of primary surgery, and ovarian (Krukenberg) metastases). Both the French ProphyloChip (NCT01226394) and the Dutch COLOPEC (NCT02231086) randomized trials did not meet their primary endpoint (three year disease free survival and peritoneal metastasis free-survival at 18 months, respectively) although long term results are awaited [82,83]. As also in these trials a short course (30 min) high-dosed oxaliplatin HIPEC regimen was used, questions were raised on the efficacy of IP oxaliplatin using this regimen [52]. A similar study of prophylactic HIPEC by the National Cancer Institute (NCT01095523) has been withdrawn [84].

### 4.3. Pseudomyxoma Peritonei

Appendiceal mucinous neoplasms represent a rare, histologically heterogeneous entity including low-grade appendicular neoplasm (LAMN), high-grade appendicular neoplasm (HAMN) and mucinous appendicular adenocarcinoma [85,86,87]. Ruptured low grade tumors may cause progressive accumulation of mucinous ascites and result in the pseudomyxoma peritonei (PMP) syndrome, which is a clinical and radiological phenotype rather than a histopathological diagnosis [88]. Encouraging survival results have been achieved in patients with PMP using cytoreductive surgery and HIPEC [89,90,91]. A recent international registry of over 2000 patients undergoing CRS and HIPEC for PMP showed a median survival of 16.3 years and a 10 year survival of 63% [92].

Recently, an international cohort study was published including 1924 patients with PMP, investigating the outcome after CRS with or without HIPEC [93]. It was found that the addition of HIPEC after CRS was associated with a significantly better overall survival as compared to CRS alone with a 5-year overall survival of 58% versus 46.2% respectively. The addition of HIPEC did not result in more post-operative complications. Therefore, CRS and HIPEC is proposed as the standard of care in patients with low grade appendiceal neoplasms associated with PMP [94]. Nevertheless, others argued that the favorable outcome achieved in PMP results from a favorable tumor biology and complete surgical removal rather than from the addition of HIPEC [95].

### 4.4. Gastric Cancer

The risk of peritoneal metastasis in gastric cancer is approximately 40%, with almost 30% of patients presenting with peritoneal metastases at the time of diagnosis [2]. In the Far East (primarily Japan), promising results were obtained using prolonged IP taxane based chemotherapy in PM from gastric cancer PM [96,97].

Meta-analyses of small RCT’s and non-controlled trials show a potential benefit of HIPEC in gastric cancer patients with positive cytology and without extensive nodal disease [98,99]. A recent propensity score adjusted comparison of CRS alone versus CRS with HIPEC in patients with PM from gastric cancer suggests that the addition of HIPEC results in a significant improvement of recurrence-free and overall survival [99]. Randomized trials were initiated to test the efficacy of HIPEC in gastric cancer with PM in the Netherlands (PERISCOPE II, NCT03348150), France (GASTRICHIP, NCT01882933), and China (NCT02356276). The initial results of the PERISCOPE-trial aimed at dose-finding were recently published. Although the amount of serious adverse events in the trial was high (17 out of 25 patients), it was shown that HIPEC with a dose of 50 mg/m^2^ intraperitoneal docetaxel appeared to be feasible [100] Survival data from that trial are currently awaited.

### 4.5. Other Intra-Abdominal Cancers

In patients with epitheloid malignant peritoneal mesothelioma, encouraging results were observed using CRS and HIPEC. A recent systematic review of six published studies including 240 patients showed a median survival ranging from 34 to 92 months [101]. Other peritoneal malignancies that were treated with IPDD include small bowel adenocarcinoma, peritoneal sarcomatosis, and desmoplastic small round cell tumors [102,103].

## 5. Addressing Current Limitations of HIPEC: The Road to Progress

### 5.1. Development of Novel Anticancer Compounds and Carriers

The main current limitation of HIPEC is that none of the currently used drugs were developed for intraperitoneal administration. Toxic effects of chemotherapy or the carrier solution on mesothelial integrity may offset anticancer efficacy. Also, HIPEC is performed only once, and treatment duration is typically short. Data from in vitro experiments suggest, that upon exposure to hyperthermia, the observed decrease in cancer cell survival decreases exponentially with longer treatment duration [104]. Also, simulations based on experimental data show that the response of cancer cell lines to chemotherapy depends not only on cell cycle phase specificity, cell cycle time, and drug concentration, but also on treatment duration [105]. Emerging approaches to extend treatment duration are the development of nanoparticles and prolonged delivery formulations such as hydrogels and drug loaded textiles [106]. While these may not be easily administered as a hyperthermic chemoperfusion, they may be combined with external sources of hyperthermia such as radiofrequency or with photothermal activation [107,108,109,110].

### 5.2. Improved Heat Delivery Methods

Homogeneous tissue heating is impeded by insufficient and preferential fluid flow and heat sink effects. Recently, studies based on computational fluid dynamics (CFD) were used to simulate fluid flow, temperature, and drug distribution to predict the influence of location and number of catheters, flow alternations, and flow rate [111]. The results of these studies, combined with adequate thermometry methods, may allow to improve spatial homogeneity of heat and drug in the peritoneal cavity.

### 5.3. Clinically Relevant Preclinical Models

HIPEC treatment was introduced in clinical practice in the absence of solid preclinical foundations. Given the challenging results of HIPEC in CRC, there is a need for clinically relevant, reproducible, and high throughput models to study the immune and anticancer effects of HIPEC in a systematic way. Several groups have established mouse HIPEC models, which offer the advantage of antibody availability and the potential to use human cell lines [112,113]. Recent developments include the use of patient derived organoids and ‘organs on a chip’ in order to study the effects of hyperthermia combined with anticancer agents using patient derived tissue [78,114,115].

### 5.4. Elucidation of the Tumor Microenvironment and the Peritoneal Ecosystem

It is increasingly evident that the behavior and treatment response of solid tumors is largely dictated by its biophysical, cellular, and molecular environment. This environment is radically different between primary tumors and their associated PM. Therefore, unraveling of the PM cascade and understanding the PM-associated tumor microenvironment (TME) are priorities for future research [116]. Furthermore, the immune contexture of PM and the peritoneal ecosystem, and how both are affected by extensive surgery, IP chemotherapy, and hyperthermia are barely studied and need to be characterized in detail.

### 5.5. High Quality Clinical Trials

After a long period of skepticism, HIPEC has gathered significant momentum over the past years, with 121 centers offering the treatment in the US alone in November 2019 [114]. In stark contrast, only a handful of RCTs has studied the efficacy of HIPEC. The obstacles faced by surgeon initiated trials are well known: perceived lack of equipoise, lack of training in clinical trial methodology, learning curve effects, and lack of funding [115]. Although the RCT remains the gold standard, possible alternative approaches that allow to facilitate gathering evidence on HIPEC include pragmatic trials, register based trials, patient preference trials, and adaptive (Bayesian) trial designs [117].

## 6. Conclusions

There are sound theoretical arguments that favor the incorporation of HIPEC in a multimodal strategy for patients with PM. Its current place remains, however, uncertain due to the significant variability in the drugs and methods used to deliver HIPEC. Also, results from clinical trials are inconsistent. Further development of HIPEC will require a better understanding of how surgery and HIPEC affect the tumor TME and peritoneal ecosystem. In addition, the role of treatment variables such as chemoperfusion temperature, HIPEC duration, and chemotherapeutic drug(s) need to be established. At the same time, efforts should be directed to the development of novel IP compounds and delivery systems, and to the expansion of the clinical evidence from randomized trials.

## Figures and Tables

**Figure 1 cancers-13-03114-f001:**
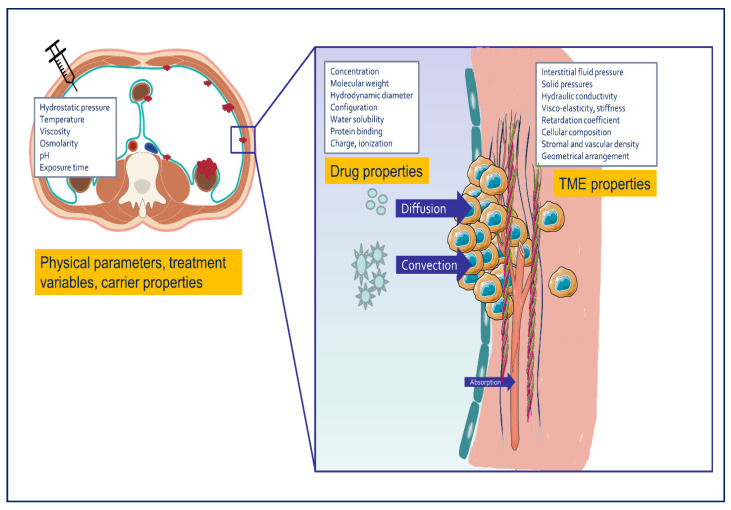
Overview of relevant mechanisms and variables that affect tissue transport after intraperitoneal drug delivery. Drug transport is driven by convection (pressure gradient) and by diffusion (concentration gradient). The ratio of convective/diffusive transport is larger for large or nanosized compounds.

**Figure 2 cancers-13-03114-f002:**
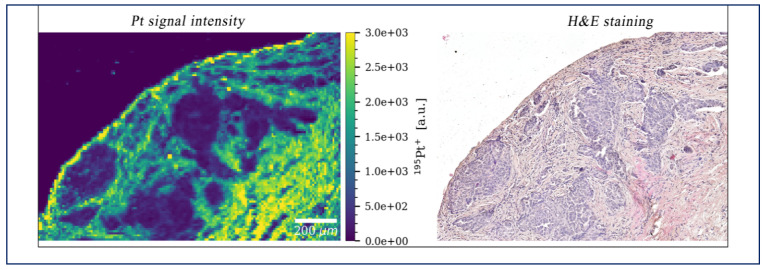
Platinum (Pt) penetration after HIPEC using cisplatin in a patient with peritoneal metastases from ovarian cancer. When comparing Pt penetration with histology, it is obvious that Pt penetrates the stroma much more efficiently compared to the nests of cancer cells.

**Table 1 cancers-13-03114-t001:** Overview of randomized trials comparing surgery combined with HIPEC versus surgery alone.

Tumor	Study, Year	Inclusion	Primary Endpoint	Treatment and N Randomized	Results	95% CI of Effect and *P* Value
Colorectal cancer	Verwaal [58] (2003, updated 2008)	Histologically proven PM, age <71 yrs, no distant metastasis	Disease specific survival	Chemotherapy alone (5-FU-LV) *N* = 51	12.6 m	*P* = 0.028
				CRS and HIPEC (MMC, 90 min) *N* = 54	22.2 m	
	Prodige 7 (2021) [15]	Histologically proven PM, PCI ≤25	Overall survival	CRS *N* = 132	41.2 m	HR 0.63–1.58, *P* = 0.99
				CRS and HIPEC (OX, 30 min) *N* = 133	41.7 m	
	COLOPEC (2019) [59]	Clinical or pathological T_4_N_0–2_M_0_-or perforated colon cancer	Peritoneal metastasis free survival at 18 months	Adjuvant HIPEC (OX, 30 min) and adjuvant chemotherapy *N* = 102	80.9%	*P* = 0.28
				Adjuvant chemotherapy *N* = 102	76.2%	
	PROPHYLOCHIP (2020) [60]	Synchronous and resected PM, resected ovarian metastases, perforated tumor	Disease free survival	Adjuvant chemotherapy and HIPEC (OX ± IRI, 30 min) *N* = 75	44%	HR 0.61–1.56, *P* = 0.82
				Adjuvant chemotherapy *N* = 75	53%	
	Rovers (2021) [61]	Histologically proven isolated resectable PM	% complete CRS/% Clavien-Dindo ≥ grade 3 morbidity	Perioperative chemotherapy and CRS-HIPEC (MMC, 90 min or OX, 30 min) *N* = 40	89%/22%	RR 0.88-1.23, *P* = 0.74/0.31–1.37, *P* = 0.25
				CRS and HIPEC alone *N* = 40	86%/33%	
Ovarian cancer	Spiliotis (2015) [62]	Recurrent EOC	Overall survival	CRS and HIPEC (CIS or DOX with PTX or MMC, 60 min) *N* = 60	26.7 m	*P* = 0.006
				CRS alone *N* = 60	13.4 m	
	OVHIPEC (2018) [63]	EOC with at least stable disease after three cycles of carboplatin–PTX	Recurrence free survival	Interval CRS and HIPEC (CIS, 90 min) *N* = 122	14.2 m	HR 0.50–0.87, *P* = 0.003
				Interval CRS alone *N* = 123	10.7 m	
	Zivanovic (2021) [64]	Recurrent EOC	Proportion free of progression at 24 months (‘pick the winner’)	CRS and HIPEC (Carboplatin, 90 min) followed by 5 cycles of Carboplatin based IV chemotherapy *N* = 49	16.3%	Not applicable (no winner determined)
				CRS alone followed by 6 cycles of Carboplatin based IV chemotherapy *N* = 49	24.5%	

*Abbreviations*: PM, peritoneal metastases; CRS, cytoreductive surgery; HIPEC, hyperthermic intraperitoneal chemoperfusion; CI, confidence interval; HR, hazard ratio; RR, risk ratio; PCI, peritoneal cancer index; OX, oxaliplatin; IRI, irinotecan; CIS, cisplatin; PTX, paclitaxel; MMC, mitomycin C; DOX, doxorubicin; EOC, epithelial ovarian cancer.

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
