# Peer review of "Hyperthermic Intraperitoneal Chemotherapy: A Critical Review"

_cancers, 2021, doi:10.3390/cancers13133114_

Round 1

Reviewer 1 Report

In this article, the authors provide a critical overview of the theoretical concept and preclinical and clinical study results and propose a framework to better define the role of HIPEC in the treatment of peritoneal malignancies. The manuscript is straightforward, well written, and concise and has clear results within the scope of a review article. Definitely deserves to be published and is a valuable contribution to the “cancersjournal. Some minor flaws need to be addressed before publication.

Minor points:

[1] “1. Introduction”, Page 2/16, Lines 81-83:

Nevertheless, the efficacy and safety of HIPEC remain debated and hamper the universal acceptance by the oncology community.”.

At that point, please do report that HIPEC does not increase the mortality and morbidity compared to cytoreductive surgery alone in ovarian cancer. The latest National Comprehensive Cancer Network (NCCN) guidelines recommend HIPEC at interval cytoreduction. Overall, HIPEC has been better investigated in the recurrent setting, resulted in improved survival.

Recommended reference: Boussios S, et al. Ovarian cancer: state of the art and perspectives of clinical research. Ann Transl Med. 2020;8(24):1702.

[2] “1. Introduction”, Page 2/16, Lines 83-86:

Proponents will argue that the addition of HIPEC was recently shown to prolong survival in ovarian cancer in a randomized clinical trial (RCT) but criticism was undoubtedly fueled by negative results of RCTs in patients with colorectal cancer (CRC) PM.15”.

Moreover, whether HIPEC is of benefit or cytoreductive surgery can be omitted in the subset of patients with serous primary peritoneal carcinoma (SPPC), should be addressed in specifically designed trials. SPPC is not commonly distinguished as a distinct clinical entity for clinical trial inclusion and has been enrolled in ovarian cancer trials. The better understanding of the biology of SPPC permits a strict disease definition that creates a common standard diagnostic workup and a homogeneous patient population.

Recommended reference: Rassy E, et al. Narrative review on serous primary peritoneal carcinoma of unknown primary site: four questions to be answered. Ann Transl Med. 2020;8(24):1709.

[3] “2.3. Use of hyperthermia”, Page 5/16, Lines 182-184:

Other drugs showing increased tumor penetration when combined with hyperthermia include carboplatin, oxaliplatin, and doxorubicin.33,34”.

In addition, hyperthermia may also diminish the systemic toxicity of some drugs, such as doxorubicin and cyclophosphamide by increasing their alkylation and/or excretion.

Recommended reference: Boussios S, et al. Malignant peritoneal mesothelioma: clinical aspects, and therapeutic perspectives. Ann Gastroenterol. 2018 Nov-Dec;31(6):659-669.

[4] “4.1. Ovarian cancer”, Page 7/16, General comment:

I would recommend at the end of this section to raise the main question emerging further research. Which is the best setting to perform HIPEC in ovarian cancer. Is it the upfront setting or the relapsed disease?

[5] General comment (1):

The paper could be substantially improved from good visualizations, since they would make the content more clear. For that reason, please, create a table summarizing the most important clinical trials at least for ovarian and colorectal cancer. This is key element for an article to be attractive for the readers.

[6] General comment (2):

A workflow diagram for the study would be of benefit for the readers.

Reviewer 2 Report

This critical review is concise, but does discuss and include all main issues relevant for the present status and projected future directions of HIPEC. I therefore advise acceptance after the authors have resolved the following issues:

Page 1, Abstract, line 27/28: you state ‘results from clinical trials have been equivocal’, surely you mean ‘results from clinical trials have been unequivocal’ in this sentence?

Page 1, Introduction, line 36-38: you are only mentioning ovarium, colorectum, lung, melanoma and breast cancer as sites metastasizing to the peritoneum, but you also mention pancreas and gastric in the paper, should they not be mentioned here as well?

Page 5, line 200-202: you claim there have been no studies comparing normothermic and hyperthermic HIPEC, but actually there is a three armed study that did so for normothermic MMC HIPEC versus hyperthermic MMC HIPEC and they found that normothermic HIPEC offered no benefit over the control arm (139 T2–4 gastric cancer patients, CRS + HIPEC: 30 mg MMC+ 300 mg cisp/6-8lt) with a 5 yr OS of 42% for the control arm, 43% for HIPEC at 37C, 61% for HIPEC at 42-43C.

Ref: Y. Yonemura, X. de Aretxabala, T. Fujimura, S. Fushida, K. Katayama, E. Bandou, K. Sugiyama, T. Kawamura, K. Kinoshita, Y. Endou, T. Sasaki, Intraoperative chemohyperthermic peritoneal perfusion as an adjuvant to gastric cancer: final results of a randomized controlled study, Hepatogastroenterology 48 (2001) 1776–1782.

Page 8, line 328: you state that the two prophylactic colorectal HIPEC trials ‘were negative’, one might argue that in at least one case outcome is still undecided due to too short FU, so I would reformulate subtly to ‘did not prove a benefit’.

Page 9, line 381-383: you mention use of RF for treatment of the peritoneum, I suggest referring also to papers on dedicated RF heating of the peritoneal cavity:

Beck M, Ghadjar P, Weihrauch M, Burock S, Budach V, Nadobny J, Sehouli J, Wust P. Regional hyperthermia of the abdomen, a pilot study towards the treatment of peritoneal carcinomatosis. Radiat Oncol. 2015 Jul 30;10:157

Kok HP, Beck M, Löke DR, Helderman RFCPA, van Tienhoven G, Ghadjar P, Wust P, Crezee H. Locoregional peritoneal hyperthermia to enhance the effectiveness of chemotherapy in patients with peritoneal carcinomatosis: a simulation study comparing different locoregional heating systems. Int J Hyperthermia. 2020;37(1):76-88

Reviewer 3 Report

The authors review a large number of publications concerning the interesting field of HIPEC. The main disadvantage is that this overview yields unsatisfying results. The studies are too inhomogeneous and contradictory – as the authors concede, too.

Therefore, there is no clear valuable message in the article but the message that more has to be done in this field of hyperthermia. However, is this enough for publication? If the authors could find additional clear consistent conclusions in the different studies, it would increase the value of the article.

Further, the reached temperature and the duration of the treatment are very important aspects in hyperthermia. The authors report the treatment time of only a few studies and the temperature of (almost) none. Is there no better information about these important values? Maybe, they could help to explain the inconsistent results.

Round 2

Reviewer 2 Report

the authors have revised their manuscript in a satisfactory fashion, I recommend acceptance

Author Response

Thank you for approving our submission.

Reviewer 3 Report

The authors could really improve the article for instance by including hyperthermia temperatures, hyperthermia durations and table 1. Thus, it is now noticeable that in the randomized studies (with and without HIPEC) the additional HIPEC yields a significant advantage if the hyperthermia duration is long enough. This interesting and important result should be included e.g. in the conclusion by the authors.

Because the duration of hyperthermia is very important (see above), the authors should include this aspect in chapter 5 by an own subchapter or at least in 5.2.

Some minor remarks:

  • In line 59: there are two-times “mixture of”.
  • In line 230: phase instead of fase?
  • In line 373: CRS instead of HIPEC?
  • In line 447: some instead of sound?
  • Only an aesthetic aspect: In table 1 the sequence of the cancer types is vice versa to the sequence in the text.

Author Response

The authors could really improve the article for instance by including hyperthermia temperatures, hyperthermia durations and table 1. Thus, it is now noticeable that in the randomized studies (with and without HIPEC) the additional HIPEC yields a significant advantage if the hyperthermia duration is long enough. This interesting and important result should be included e.g. in the conclusion by the authors.

We have added this to the conclusion.

Because the duration of hyperthermia is very important (see above), the authors should include this aspect in chapter 5 by an own subchapter or at least in 5.2.

We have added additional information on the importance of treatment duration under section 5.1.

Some minor remarks:

In line 59: there are two-times “mixture of”.

Thank you for pointing out this mistake. It was corrected.

In line 230: phase instead of fase?

Thank you for pointing out this mistake. It was corrected.

In line 373: CRS instead of HIPEC?

Thank you for pointing out this mistake. It was corrected.

In line 447: some instead of sound?

‘Sound’ is correct here, meaning ‘based on good judgment’.

Only an aesthetic aspect: In table 1 the sequence of the cancer types is vice versa to the sequence in the text.

We agree, and this was corrected.